# X-ray rheography uncovers planar granular flows despite non-planar walls

James Baker [1], François Guillard [1], Benjy Marks [1] & Itai Einav [1]

Extremely useful techniques exist to observe the interior of deforming opaque materials, but these methods either require that the sample is replaced with a model material or that the motion is stopped intermittently. For example, X-ray computed tomography cannot measure the continuous flow of materials due to the significant scanning time required for density reconstruction. Here we resolve this technological gap with an alternative X-ray method that does not require such tomographs. Instead our approach uses correlation analysis of successive high-speed radiographs from just three directions to directly reconstruct three-dimensional velocities. When demonstrated on a steady granular system, we discover a compressible flow field that has planar streamlines despite curved confining boundaries, in surprising contrast to Newtonian fluids. More generally, our new X-ray technique can be applied using synchronous source/detector pairs to investigate transient phenomena in various soft matter such as biological tissues, geomaterials and foams.

---

[1] School of Civil Engineering, The University of Sydney, Sydney, NSW 2006, Australia. Correspondence and requests for materials should be addressed to I.E. (email: itai.einav@sydney.edu.au)

When strolling along the beach, our footprints tell us that the sand under the surface must have moved but not precisely where or how. Similar internal deformations frequently occur in particulate media such as snow, cereals and pharmaceutical powders, motivating numerous scientific investigations into granular flow. The laboratory paradigm closely resembles the beach scenario: we cannot see inside these typically opaque materials and thus mostly make inferences from observations at the boundaries or from numerical simulations. Despite this difficulty, experiments have revealed granular media to exhibit rich flow behaviour[1]. While these granular phenomena frequently have analogues in classical fluids, for example convection rolls[2] and hydraulic jumps[3], the underlying physics is distinct and often poorly understood. Other effects, such as localised shear bands[4] and particle size segregation[5], are unique to granular systems and thus offer their own challenges. Much insight can be gained using sophisticated image analysis at the flow surfaces to accurately determine strain and velocity fields[6–9], but the measurements are not necessarily representative of the bulk[10].

Numerous techniques have been employed to resolve the interior of the flow, including X-ray computed tomography (X-ray CT)[11–14], positron emission particle tracking[15], magnetic resonance imaging[16,17], ultrasonic imaging[18], and refractive index-matched scanning[19,20]. Each has their advantages depending on the specific flow regime to be investigated and the focus of that investigation. However, there are also many drawbacks such as poor spatial or temporal resolutions, high costs, or obtrusiveness, and there is yet to arise a single stand-out method that is favourable for all types of granular flow. X-ray technologies are arguably the most promising in this respect, with micro- and nano-CT machines being capable of reconstructing internal density fields in great detail without interfering with the sample. Such reconstructions require radiographs from many scanning angles, which is usually achieved with a single X-ray source by rotating either the source or specimen[14]. However, such a process takes time, meaning that CT can typically only be used to investigate quasi-static deformations. Alternatively, given an initial three-dimensional (3D) density map, just two projections can be used to deduce subsequent material displacements[21], significantly shortening the relevant timescales. However, this method still requires a single full CT scan and loses accuracy once particles have moved significantly from their reference position.

Recent technological improvements have made it possible to collect successive radiographs from a single detector panel at frame rates comparable to continuous flow timescales. This high-speed radiography has been exploited in both Newtonian fluid[22,23] and granular flows[24] by applying the image correlation methods from particle image velocimetry (PIV) to pairs of radiographs, taking the cross-correlation peaks to deduce the most likely displacement between images. Owing to the projective nature of radiography, this peak gives a measure of the beam-averaged velocity, which is the modal value taken over all positions in the path of the X-ray beam. Such an approach eliminates boundary-layer effects that can cause misleading optical wall measurements[10]. The same radiographs can also be used to obtain the evolving size and orientation of particles through the use of Fourier transform-based methods[24], again in a two-dimensional (2D), beam-averaged sense.

When taken in isolation, a single set of radiographs cannot be used to recover out-of-plane velocities. However, by considering whole cross-correlation functions as opposed to only the peak locations, one can infer additional details about the out-of-plane variation, for example by assuming these cross-correlations represent the probability distribution of velocities through the beam[25]. Under certain flow assumptions, this extra information

alone is sufficient to reconstruct full 3D fields, but in general it must be combined with results from other scanning angles to build a complete picture. Such an approach has been successfully applied to classical fluids using nine projection angles[26] and, by exploiting incompressibility, extended to simultaneously measure flow geometries and internal velocities with as few as three projections[27]. However, these methods rely on a rigid optimisation procedure that restricts the solution to pre-assumed forms of velocity fields, for example divergence-free flows. In this paper, we develop a new velocimetry method that requires no such restrictions and is thus completely general. It is applied to measure, for the first time, 3D velocity fields in granular materials, where compressibility cannot be neglected. As for the previous correlation-based methods, the new velocity reconstruction technique does not require any tomographic density maps, and we thus call it X-ray rheography to distinguish from existing X-ray CT technology. A brief outline of the approach is given below, with full details provided in the Methods section. Since validation of any new methodology is crucial, we have conducted two different validation procedures. In the first procedure, described below and in the Methods, discrete element method (DEM) simulations of the specific experimental set-up are computed, which are used to generate artificial radiographs from the data. Our new rheographic technique is then applied to these images, with the results being compared to the coarse-grained numerical flow fields as well as the experimental reconstructions. In the second procedure, we have also generated radiographs, yet from a simple known flow field, as this allows a more precise quantification of the errors introduced at each stage in the reconstruction process. Details of this second validation method are provided as Supplementary Methods.

## Results

**Rheographic process.** The experimental set-up is shown in Fig. 1, where a confined, steady-state flow is established inside a cylinder using a constant velocity conveyor belt to shear a granular material (either spherical glass beads or elongated pearl barley) from below. In such time-independent flows, a single X-ray source/detector pair can be repositioned at different angles around the sample while the underlying velocity field remains constant. Here we use three mutually perpendicular orientations along the directions $x$, $y$, and $z$, corresponding to the basal shear, cross-belt, and vertical directions, respectively. From each angle, successive radiographs are recorded at a rate of 30 fps (frames per second) during flow, as shown in Fig. 2 and in Supplementary Movie 1.

For validation purposes, we also simulate the experimental regime for the spherical particles using DEM computations. These simulations are used to generate a series of artificial radiographs (Fig. 2a–c and Supplementary Movie 1), which bear a strong resemblance to the real radiographs for glass beads (Fig. 2d–f). The artificial images are slightly more exposed, suggesting that a lower theoretical X-ray intensity and/or different attenuation coefficient could be used to generate more realistic radiographs. However, their primary purpose is to compare the results of the rheographic process (using artificial images) with the DEM velocity field, and so exact agreement between our forward projections and the experimental radiographs is not critical. Furthermore, for the subsequent correlation analysis the images are normalised by the average intensity, meaning that only the intensity fluctuations are of interest.

Next, both the real experimental and artificially generated radiographs are analysed by first dividing into 32 × 32 interrogation windows. For a given window, the auto- and cross-correlation functions are computed and, by solving a deconvolution inverse

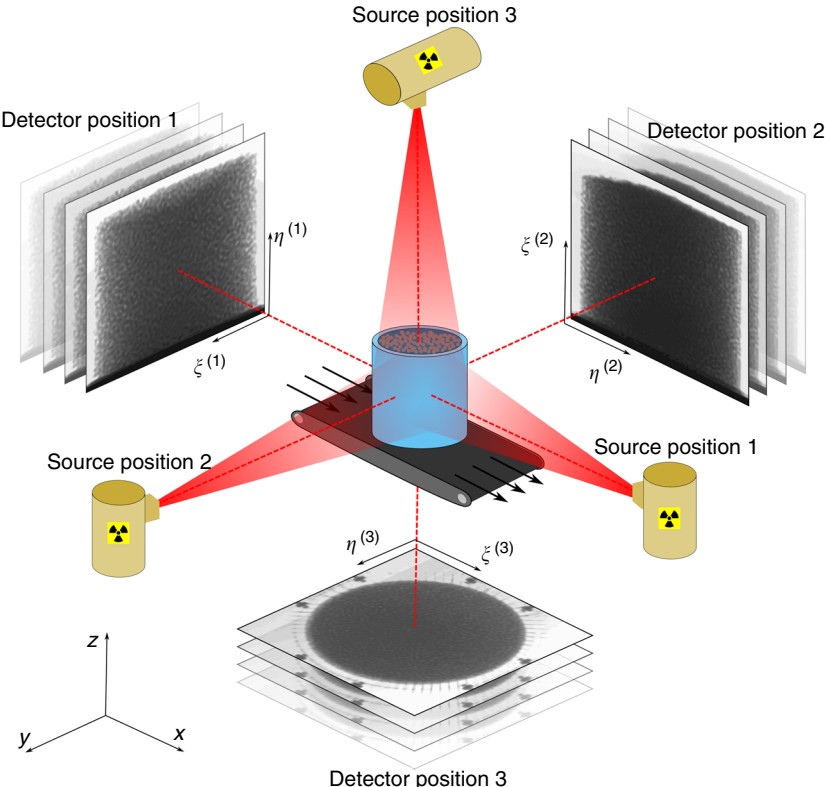

**Fig. 1** Schematic diagram of the experimental set-up. Confined granular material lies inside the blue cylinder, which is sheared from below in the direction indicated. The three X-ray source positions are shown in yellow, with red indicating the approximate extent of beams, and greyscale images representing radiographic acquisition at the corresponding detector positions. Also shown for reference are the global ($x$, $y$, $z$) and local imaging coordinates ($\xi$, $\eta$)

problem, used to calculate the probability density functions (PDFs) of in-plane displacements in that window (Fig. 3a). These PDFs give the distribution, along the path of the X-ray beam, of the in-plane velocities, which we discretise by taking equally spaced percentiles (Fig. 3b) into vectors of candidate velocities, considering the two planar components separately. For each window and component this tells us the different displacements perpendicular to the beam that occur through the direction of the beam, but does not give the relative out-of-plane position of each such component. An orthogonal scanning angle provides the additional information required to reconstruct a 3D map of the velocity component in plane with both detectors, by combining the two sets of candidate arrays. For example, scanning directions 1 and 2 (as shown in Fig. 1) are used for the vertical component. Given these two candidate arrays, the problem becomes a question of how best to consistently arrange the vectors in the internal grid. This may be thought of as a generalised Sudoku-style puzzle because we know the numbers that should be placed in each row and column but not how to arrange them. However, unlike a Sudoku problem, we are not given any filled-in cells to start the process, and the row and column information may not perfectly match each other. Nevertheless, it can be formulated into a simple optimisation problem, where one aims to minimise discrepancies between the two sets of observations, which we solve with a fast, constructive algorithm to find the arrangement of the two sets of candidate arrays. Taking the mean of these two vectors in each cell gives the output solution value. Note that the solution of this algorithm is not unique, being dependent on the order in which each internal grid point is calculated. This problem is overcome by computing several comparable solutions using different random paths through this grid and averaging over these values. The other

two components of the velocity are reconstructed in the same manner by repeating with other pairs of scanning directions.

**Rheographic results**. The experimental velocity fields obtained by this rheographic method are shown in Fig. 4a, b, d, e, where it can be seen that both glass beads and barley display similar flow kinematics. The rough conveyor belt induces forward motion in a narrow layer of grains near the base, which rise as they approach the front wall before retreating backwards, eventually falling back towards the base. This process leads to the formation of vortex-like rotational structures, where the streamlines are approximately planar, indicating a 2D primary flow (Fig. 4a, b). This is somewhat surprising, given the curvature of the walls of the cylindrical geometry, but is also observed in the coarse-grained DEM velocity fields (Fig. 5a) as well as the rheographic velocities from artificial radiographs (Fig. 5b). Figure 5 therefore provides both confirmation of this finding, as well as validation for the rheographic method.

Even though each velocity streamline is largely restricted to a single plane, the field of velocity magnitudes is genuinely 3D (Fig. 4d, e). To understand how this planar motion gives rise to a 3D field, consider that individual particles, despite being confined to 2D trajectories, are free to accelerate and decelerate as they pass through different positions. If two particles on neighbouring streamlines adjust their speed differently to each other, then the resulting flow map will take on a 3D nature. This effect is only possible if the flow in question is compressible, and Fig. 6 confirms that the divergence fields are indeed non-zero. While there are areas of weak compression and dilation in the upper regions of the flow (Fig. 6a, b), the effect is most striking towards

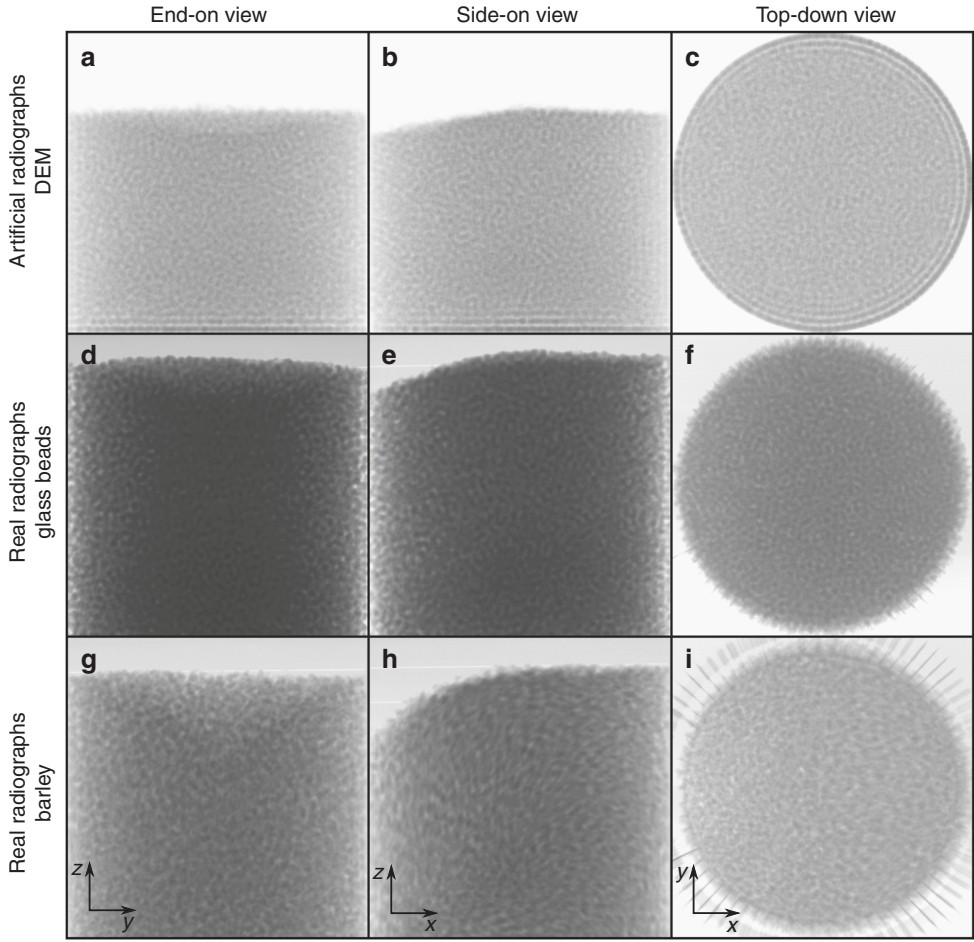

**Fig. 2** Artificial and real radiographs. Snapshot of **a**–**c** artificial radiographs generated from DEM data, **d**–**f** real radiographs obtained from glass bead experiments, and **g**–**i** real radiographs from pearl barley experiments, with the three orthogonal viewing directions shown in each case. Supplementary Movie 1 is available online showing the time evolution of all radiographs

the base, with Fig. 6c, d showing regions of expansion as material begins to rise from the bottom, and compressive regions as it returns back to the base. In contrast, filling the same geometry with an incompressible Newtonian fluid (Fig. 4c, f) leads to a drastically different result. Steady laminar flow simulations using the Fluent module of ANSYS Academic Research Mechanical, Release 17.1 (see Methods), show that the flow field is again strongly 3D but, since expansion and contraction do not occur, the streamlines are now also strongly 3D. Compressibility effects are often neglected for simplicity in granular constitutive laws[28] but may be necessary for well-posed and physically relevant models[29,30]. Our observations provide experimental justification for their inclusion, as a consequence of the measured flow field. The physical explanation for why this flow field is so different to a viscous fluid remains unclear but may be due to the frictional nature of grains and the non-locality arising from the intrinsic particle length scale[31]. The non-slip boundary condition for Newtonian fluids could also be playing an important role.

## Discussion

While the observed planar primary flow is a robust feature for both glass beads and barley, these experiments also highlight how particle shape influences the bulk flow field. Comparing the results for the two species across the whole domain, we find that the elongated barley grains travel, on average, 30% faster than their spherical counterparts. This occurs even though the mean

axis length of barley particles (3.8 mm) is larger than the mean diameter of the glass beads (3.2 mm), a fact typically associated with greater effective friction in granular rheologies[28]. The flow radiographs (see Fig. 2 and Supplementary Movie 1) offer one explanation, showing that the barley particles have a tendency to align their major axis with the primary flow direction, which is supported by recent models[32]. This means that the effective contact friction between grains is actually between their minor axes (2.1 mm), resulting in enhanced overall flowability compared to glass beads. Figure 4d, e show that the most significant differences are in the fastest flowing regions where grains rise from the base or retreat down the free surface, which is observed to be steeper for the elongated particles. Note that the distinctions could also be attributed to different interparticle surface friction for the two species.

Studying shape effects in granular flow has been historically challenging, largely due to the difficulties in characterising experimental results and conducting DEM simulations for verification (unlike for spherical particles, as used for validation here). However, recent kinematic models[32] provide a simple analytical framework, and Fourier-based image analysis of high-speed X-rays have been used to give experimental fabric measurements of particle size, shape, and alignment[24]. Although these have currently only been applied in a 2D, beam-averaged sense, it may be possible to relate the energy spectrums obtained from the Fourier analysis to PDFs of the relevant fields, allowing similar

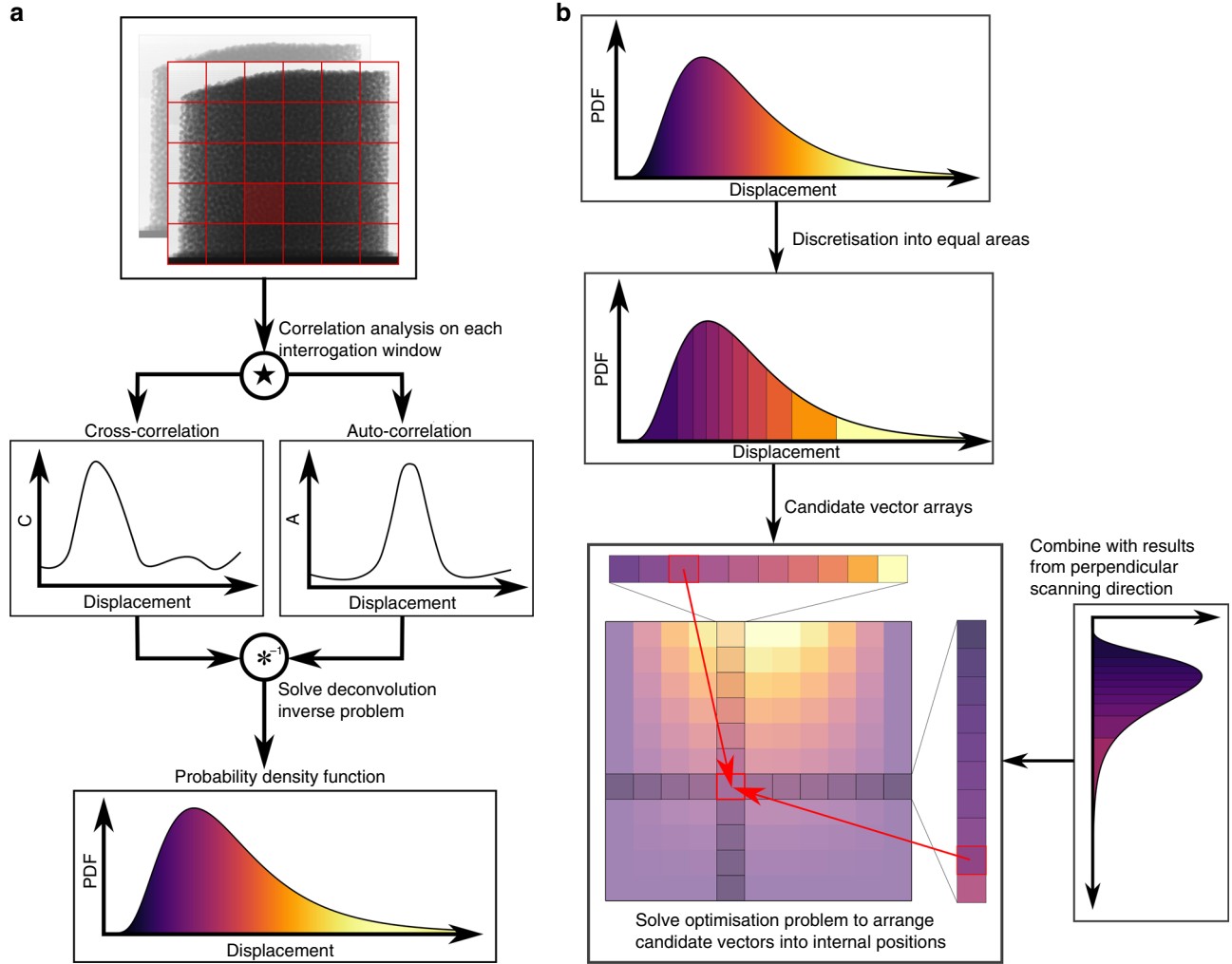

**Fig. 3** Velocity reconstruction process. **a** An initial pair of radiographs (top) is divided into interrogation windows (red grid), with image correlation analysis then conducted on each window (middle) to extract the probability density function (PDF) of displacements in that window (bottom). **b** Displacement PDFs are discretised into candidate vectors and results are combined with orthogonal direction to arrange candidate arrays into the internal grid positions

3D reconstructions to the velocity field. Such unobtrusive tools will advance the study of non-spherical grains and enable investigation of other inhomogeneous granular phenomena, for example the coupling between particle size segregation and bulk velocity fields when grains of different shapes and sizes are present.

The X-ray rheographic technology presented in this paper can be further developed in future using three synchronous pairs to measure transient flows, an enticing prospect for those working on amorphous soft matter systems such as foams, suspensions, geomaterials, or biological fluid flows.

## Methods

**Experimental set-up and image collection**. A 3D-printed PLA cylinder of height 100 mm and inner radius 50 mm is held in a fixed position 0.5 mm above a rough rubber conveyor belt (typical belt asperities of size 3 mm) to form an internal cavity that is filled with granular material. We carry out two different experiments, the first using sub-spherical glass beads of diameter $3.2 \pm 0.3$ mm and the second using ellipsoidal pearl barley with principal axes of length $6.1 \pm 0.4$ mm, $3.1 \pm 0.5$ mm and $2.1 \pm 0.2$ mm. In both cases, the same volume of material (1 litre at random loose packing) is used, and the belt is driven at a fixed velocity of $8.3$ mm s$^{-1}$. The gap between belt and cylinder is large enough for the belt to move freely but small enough for particles to remain confined to the inside of the cylinder. This set-up allows a steady-state granular flow to be established, which can therefore be investigated by repositioning a single X-ray source (Spellman XRV generator with Varian NDI-225-21 stationary anode tube) and detector panel (PaxScan 2520DX).

We use three mutually perpendicular directions with the normal to the detector panel pointing along each axis of a global orthogonal coordinate system $xyz$, where the positive $x$ axis is in the direction of belt motion, $y$ is in the cross-belt direction, and $z$ points vertically upwards. The experimental set-up is designed so that motors and aluminium supports do not obstruct the field of view from any of these positions, with approximate source–detector distance of 2 m to minimise non-parallel beam effects. From an external control room, we set the source to emit radiation at a maximum energy of 150 keV and intensity of 5 mA and continuously record $960 \times 768$ px$^2$ (16-bit) radiographic images at a rate of 30 fps. For each direction, local imaging coordinates $(\xi, \eta)$ are used to refer to the two in-plane directions, which are related to the global axes via

$$
\begin{aligned}
\left(\xi^{(1)}, \eta^{(1)}\right) &\mapsto (y, z), \\
\left(\xi^{(2)}, \eta^{(2)}\right) &\mapsto (z, x), \\
\left(\xi^{(3)}, \eta^{(3)}\right) &\mapsto (x, y),
\end{aligned}
\tag{1}
$$

as indicated in Fig. 1.

**Image correlation and convolution analysis**. Images are preprocessed by dividing through by the average intensity, calculated from 5000 radiographs taken from the same scanning direction during flow. This highlights the intensity fluctuations, or equivalently density fluctuations, from the baseline values, which are significant in our system due to the relatively large particle size compared to the overall system. With increasing numbers of particles, the radiographs will become more homogeneous and the subsequent analysis will be less effective.

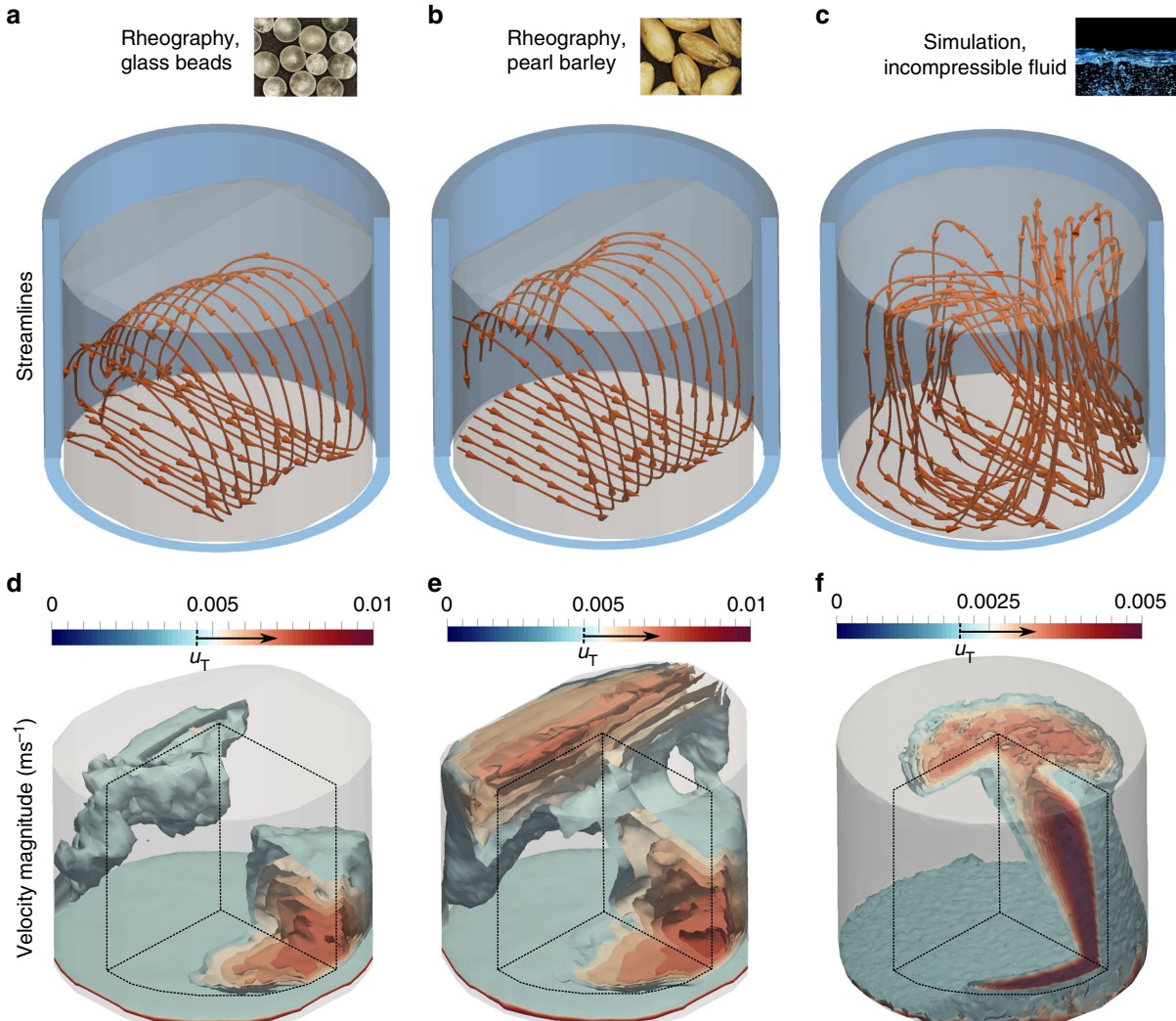

**Fig. 4** X-ray rheography and incompressible fluid simulation results. Experimental results for glass beads (**a**, **d**) and pearl barley (**b**, **e**), contrasted with the output of an incompressible Newtonian fluid simulation using ANSYS® Fluent (**c**, **f**). Streamlines of velocity $\mathbf{u} = (u, v, w)$ shown in orange on **a**–**c**, with approximate extent of flowing material in translucent grey and cut cylindrical geometry in blue. **d**–**f** show three-dimensional contour surfaces of the velocity magnitude $|\mathbf{u}| = (u^2 + v^2 + w^2)^{1/2}$ for values above a threshold $u_T = 0.0045$ ms$^{-1}$ (**d**, **e**) and $u_T = 0.002$ ms$^{-1}$ (**f**). Contours have been cut (planes shown by dashed lines) to ease visualisation, and approximate extent of flowing material shown in translucent grey

The processed images are then subdivided into interrogation windows of size $128 \times 128$ px$^2$, with an overlap of 75%. Note that such an overlap is only used in the first stage of the reconstruction process, where it is employed, as for regular PIV, to improve spatial resolution and accuracy. In the later stages, we neglect the overlapping nature of the windows, considering all information to be associated with the cell centre only. It may be interesting in future to carry the overlapping cells through the whole process, formulating new discretisation and optimisation procedures that take this extra detail into account.

For a given successive pair of processed images with intensities $I_1$ and $I_2$, we then calculate, in each interrogation window, the 2D auto-correlation ($A$) and cross-correlation ($C$) functions

$$A(m, n) = \sum_p \sum_q \frac{(I_1(m+p, n+q) - \bar{I}_1)(I_1(p, q) - \bar{I}_1)}{\sigma_1^2}, \quad (2)$$

$$C(m, n) = \sum_p \sum_q \frac{(I_1(m+p, n+q) - \bar{I}_1)(I_2(p, q) - \bar{I}_2)}{\sigma_1 \sigma_2}, \quad (3)$$

with $m, n$ representing discrete pixel displacements and $p, q$ pixel locations in the two in-plane directions. In the above, $\bar{I}_1, \bar{I}_2$ and $\sigma_1, \sigma_2$ denote the mean and standard deviation of the intensities taken over each window. To reduce computational time, these correlation functions are made one-dimensional by averaging over each dimension in turn, which gives two sets

of functions $A^\xi(m)$, $A^\eta(n)$, $C^\xi(m)$, $C^\eta(n)$, corresponding to displacements in the two in-plane directions. Such a one-dimensionalisation has previously been shown to not significantly influence results[27]. At the recording frame rate of 30 fps, a typical displacement is only an equivalent distance of a few detector panel pixels or less. To increase the spatial resolution, we therefore correlate sliding pairs of images registered 10 frames apart, i.e. image 1 and 11, 2 and 12, and so on. The one-dimensional auto- and cross-correlation functions are then averaged over 100 such pairs to minimise the noise present from false matches of particles.

To proceed further requires an understanding of how the measured correlation functions relate to the actual material displacements in each interrogation window. This is given by the convolution equation[26]

$$C = A * f, \quad (4)$$

where $f$ represents PDF of displacements in that window. The superscripts ($\xi, \eta$) have been dropped for simplicity but the correlation functions $C$ and $A$ and PDF $f$ in Eq. (4) are assumed one dimensional. The discrete convolution operator '*' is defined as

$$C(m) = \sum_i A(m - i) f(i). \quad (5)$$

Since both $C$ and $A$ can be measured, the one-dimensional PDF $f$ in each interrogation window can be calculated from Eq. (4) by conducting a

deconvolution. If the relationship is exact, this can be carried out as straightforward division in the Fourier domain. However, there remains a small amount of noise present, even after correlation averaging, which gets unboundedly amplified during Fourier division. This is an indication of mathematical ill-posedness, and thus the problem needs to be regularised. We employ a Tikhonov-type scheme by choosing $f$ to minimise the non-linear function

$$T(f) = \sum_m (C(m) - (A * f)(m))^2 + \alpha \sum_m (f(m) - f(m+1))^p, \quad (6)$$

subject to the PDF constraints $f \geq 0$ and $\int f = 1$. The regularisation parameters used are $\alpha = 0.01$ and $p = 2$, and Eq. (6) is solved with *Matlab*'s *fmincon* function. To reduce noise, this process is carried out using 20 sets of average auto- and cross-correlation functions in each window (from different moments in time), and the final function $f$ is taken to be the mean of these deconvolutions. Given $N^2$ interrogation windows for a single scanning direction, with each window associated with two distributions of in-plane velocity components, a single scanning direction therefore gives two $N \times N$ arrays of PDFs.

**Internal velocity reconstruction**. To reconstruct the internal velocities, each one-dimensional PDF is used to calculate a series of discrete displacement vectors by

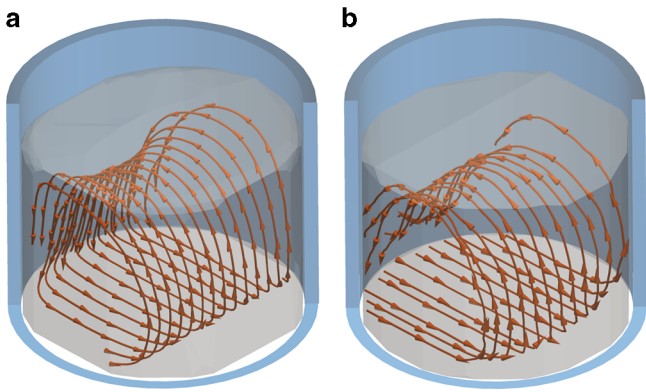

**Fig. 5** Results of validating DEM simulations. **a** shows the exact velocity streamlines, obtained directly by coarse-graining DEM results, whereas **b** show the rheographic velocity streamlines from the artificial radiographs. The streamlines are shown with orange arrows, with extent of granular material inside cut-away blue cylinder represented by transparent grey region

taking equally spaced percentiles. Specifically, a PDF $f$ is split into $N$ displacements $U_1, U_2, \ldots U_N$, where the $U_i$'s satisfy

$$\int_{U_{\min}}^{U_i} f(U) \mathrm{d}U = \frac{i}{N+1}, \quad (7)$$

for $i = 1 \ldots N$, and $U_{\min}$ is the minimum displacement, taken as minus one half of the interrogation window size. For each angle, this process gives 3D candidate arrays of size $N \times N \times N$ for the two in-plane displacement components. These constitute the displacements at different out-of-plane positions but, when taken in isolation, provide insufficient information about precisely where each displacement takes place along the beam. Taking one velocity component at a time and combining with the candidate arrays from a second scanning direction provides this extra information. Specifically, for one velocity component the discretisation process produces $2N^3$ pieces of data that are used to reconstruct the $N^3$ unknowns. For all three velocity components, our method produces $6N^3$ pieces of data that are used to reconstruct $3N^3$ unknowns. To illustrate exactly how this is done, assume that one wishes to reconstruct the vertical velocity component at a given vertical position (other positions can be calculated independently) and internal positions $(x, y)$. From the $\eta$ in-plane displacements from source/detector position 1, the candidate matrix entries $U_{i,k}^{(1)}$ are obtained, where $i$ refers to the $x$-position and $k$ indexes the discretisation of the PDF, as described above. Similarly, the $\xi$ component from position 2 gives the candidate matrix $U_{j,l}^{(2)}$, where $j$ now refers to the $y$ position and $l$ indexes the discretisation. To reconstruct the full displacements, the problem arises as to how to arrange these candidate arrays in the internal grid, i.e. how to choose the indices $k(i, j)$ and $l(i, j)$. This bears some resemblance to a Sudoku style puzzle, since we know the vectors that should be placed in each row and column but not exactly where to place them. However, unlike a Sudoku problem, the two sets of observations may not perfectly match each other. We thus seek to minimise the total matching error

$$E = \sum_{i,j} \varepsilon_{ij}, \quad (8)$$

where

$$\varepsilon_{ij} = \left| U_{ik}^{(1)} - U_{jl}^{(2)} \right|, \quad (9)$$

is the matching error at internal position $(i, j)$ in the $(x, y)$ plane. While this optimisation can be carried out directly, it is very computationally expensive, with each planar slice being an $\mathcal{O}(N!^{2N})$ operation. Instead, we use a more constructive, $\mathcal{O}(N^4)$, process. Since the problem does not have any filled-in cells to begin, we start at an arbitrary internal cell $(i, j)$. Here there are two vectors of candidate velocities, and it is simple to find the closest matching pair, i.e. to choose the indices $k(i, j), l(i, j)$ to minimise $\varepsilon_{ij}$. Much like a Sudoku problem, this initial selection of vectors removes them from the candidate arrays. The velocity in subsequent cells is calculated in the same manner, although at each step the choice

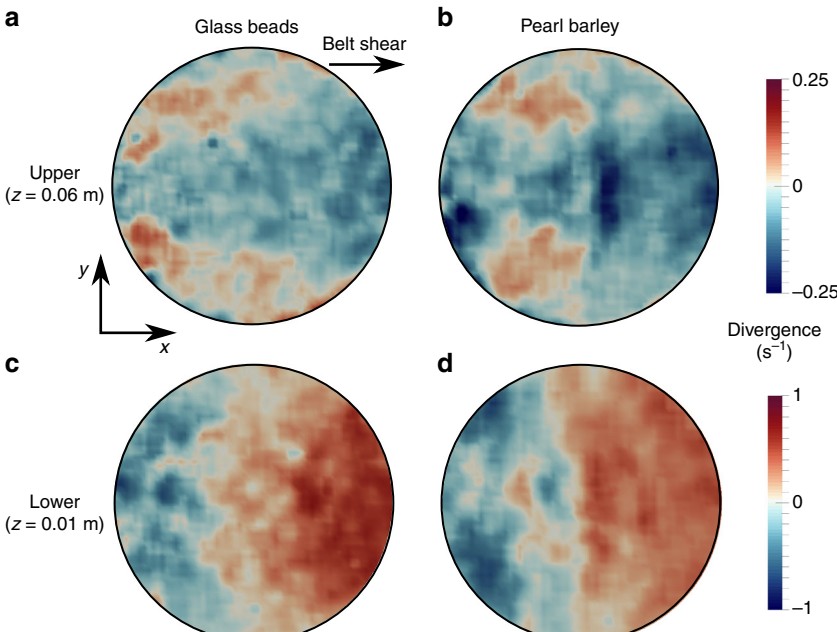

**Fig. 6** Divergence fields. Intensity maps of the measured divergence field, $\nabla \cdot \mathbf{u}$, for experiments using glass beads (**a**, **c**) and pearl barley (**b**, **d**). Horizontal slices are taken in the upper ($z = 0.06$ m, **a**, **b**) and lower ($z = 0.01$ m, **c**, **d**) regions of the flow

of possible vectors is reduced. The reconstructed velocity at each cell $(i, j)$ is then taken to be the mean of the chosen two candidate vectors

$$U_{ij} = \frac{1}{2}\left(U_{ik}^{(1)} + U_{jl}^{(2)}\right). \tag{10}$$

This process clearly depends on the order in which the cells are calculated, giving different solutions and matching errors $E$ for different paths through the internal grid. The algorithm is repeated for 5000 randomly selected paths, and the 50 solutions with the lowest value of $E$ are averaged to give the final solution, thus minimising potential non-uniqueness issues. The vertical velocity component is reconstructed at the other vertical positions in exactly the same manner, as are the other two velocity components (although using information from different detectors).

**Validation using DEM simulations**. Using MercuryDPM[33], we simulate the motion of 28,300 spherical particles with mean diameter 2.8 mm ± 10% inside a cylindrical shell of diameter 100 mm and height 100 mm. The sliding coefficient of friction between grains is taken to be 0.5, and the rolling resistance is $10^{-3}$. A rough base is generated by setting both of these coefficients to unity at the base, which is then sheared at a constant velocity 8.3 mm s$^{-1}$. The simulation is run with a time step of $\Delta t = 0.08$ s, until $t = 76$ s, although only the last 66 s are kept to allow the steady state to fully develop. Particle positions are output at the same frequency as the experimental radiographs (30 fps). These are then used to generate artificial radiographs from the three orthogonal directions, using a forward projection model described in Supplementary Methods. It is anticipated that such forward projections using DEM data can be used to further optimise the overall accuracy of the rheographic method in future.

**Newtonian laminar flow simulations**. The Newtonian fluid simulations are carried out using the Fluent module of ANSYS® Academic Research Mechanical, Release 17.1, finite element software package. The numerical domain is taken to match the physical experiment, with a cylindrical region of radius 50 mm and height 100 mm being discretised into a non-uniform mesh consisting of approximately 700,000 tetrahedral elements. Because the system has a free surface, we employ the volume of fluid method and take the lower region to be a Newtonian fluid with dynamic viscosity $\mu = 8.9 \times 10^{-4}$ Pa s and surface tension $\gamma = 0.072$ N m$^{-1}$, representing the properties of regular water. The upper region of the cylinder is taken to be a passive gas, akin to air. The interior of the cylinder is initially filled with water up to a height of 80 mm and the remainder with air, with both fluids beginning from rest. A no-slip condition is applied across the vertical domain walls, and the base is assumed to move at constant shear velocity $u_b = 10$ mm s$^{-1}$. Note that this is slower than the experimental regime but is chosen to ensure that the flow remains laminar. Indeed, the Reynolds number here is $Re = 1000$, which is significantly lower than the critical Reynolds number for turbulence in analogous 2D lid-driven cavities[34]. A pressure outlet condition is applied at the upper boundary of the computational domain. The governing equations are solved using the ANSYS least-squares cell based gradient, PRESTO pressure, third-order MUSCL momentum and compressive volume fraction spatial discretisations. For the temporal component, we use the second-order implicit time stepper, with fixed time step $\Delta t = 0.001$ s. The simulations are run for times up to $t = 100$ s to ensure that a steady state is reached.

**Code availability**. The computer codes used in this study are available from the corresponding author upon request.

## Data availability
The data that support the findings of this study are available from the corresponding author upon reasonable request.

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

## Acknowledgements
We would like to thank the Faculty of Engineering and IT for financial support through its 2014 Major Equipment Scheme and the University of Sydney for its 2017 Research Equipment Grant, both supported by the School of Civil Engineering. Our School's support is furthered thanked for its 2014 Scheme to Support High-Performing Researchers in Civil, and the Australian Research Council is acknowledged for

supporting Discovery Project DP160104310. Finally, we would like to thank Alessandro Tengattini for proposing the term 'rheography'.

## Author contributions

I.E. was responsible for the initial conception and overall management of the project. Experiments were carried out by J.B. with set-up assistance from B.M and F.G. All authors were responsible for the refinement of the rheography technique, with J.B. implementing it for the systems shown here. B.M. conducted the discrete element simulations, with J.B. generating the artificial radiographs and conducting ANSYS Fluent simulations. All authors wrote the manuscript.

## Additional information

**Competing interests:** The authors declare no competing interests.

