## [Peer Review File · Nature Communications]

Reviewers' comments:

Reviewer #1 (Remarks to the Author):

The paper introduces a technique the authors refer to Xray rheography to extract 3D velocity fields in granular flow experiments. The technique utilizes three mutually orthogonal Xray sources/detectors taking samples at a reasonably high frame rate. The resulting images are used to produce a velocity field by (i) rasterizing each of the images into a grid of interrogation windows, (ii) calculating the auto- and cross-correlation for each window, (iii) deconvolving over the correlations to produce a PDF of displacements in each window, (iv) sampling the PDF evenly to gather N candidate displacements for each window (where N is the number of out-plane slots needing of a displacement), and (v) 'syncing' the results of the different directions by placing displacements of all into positions such that all candidate displacements for all windows are used (approximately). The authors liken this last step, appropriately, to 'sudoku'.

This is a very interesting method. I do have some comments/concerns that need to be addressed before publication can be recommended.

1) For a journal like Nat Comm, novelty is important. The authors note that an analogous method has been developed already to visualize flows in fluids. What precisely are the novel elements of the current approach? Is this the very first time it has been used for granular media? Also, they say they offer "constructive, fast velocimetry" as the benefit of this approach over existing approaches. What does this mean? It sounds like the novelty is algorithmic, chiefly. In which case, more emphasis must be designated to the methods section.

2) If I understand the method correctly, a major approximation step is when the PDF's obtained from the correlations are split up into candidate displacements. This is akin to a data-coarsening. Then all these approximated displacements are sync'ed up over the whole system, which requires another approximation, whose error is attempted to be minimized in Eqs (8) and (9). The authors concede the outcome of this approach is path dependent, even if it does save orders of magnitude of compute time over a complete optimization procedure. Because of this, the authors run the algorithm many times, creating many such paths, and average the displacements from the best 50 results. How does the standard error among these 50 tests combine with the errors in the previous steps to produce error bars for this method? It would be very important to have a formula for the accuracy of this method given that approximations are made at multiple steps.

3) The validations in the supplemental materials lack a key case. The authors reproduced their experimental flow using DEM (Fig S4) and they show the velocity fields qualitatively match the outputs of the method shown in the main text. Could the authors also run a hypothetical Xray rheography of this DEM test and show the predicted velocities side-by-side with the actual? The authors do a similar analysis earlier, using a made-up exact velocity field in Fig S2, but this analysis is less convincing because the particle flow created does not necessarily follow packing constraints so it is a less realistic case.

4) What are the limitations of this method with regard to how thick a sample can be? Clearly as the sample gets thicker, the Xray intensity readouts are more opaque and potentially less useful to extract PDF's. But perhaps the authors have a clever way to address this.

Reviewer #2 (Remarks to the Author):

NCOMMS-18-24402-T presents the development of an orthogonal x-ray projection imaging based experimental method for reconstructing the internal 3D velocity field of a steady granular flow. The experimental result and the data reduction workflow are both clearly presented. The developed method does show advantages in certain aspects for the study of a flow system, although, to be fair, it does not offer the detailed information of every individual grains. Such information is available in the tomographic data and could be valuable in certain applications.

The reviewer thinks that the presented development is valuable, the proposed future directions in the technical developments and the applications are also clear. The reviewer would recommend the publication of this manuscript with minor questions listed below for the authors to consider.

1) The authors divided the projection images into a number of grids and performed correlation analysis to extract the amount of translational offset within each grid after a certain time delay. After the overall information in each grid (the displacement's probability distribution) is recovered, they numerically optimize the velocity vector distribution over the sub-grid pixels in the 3D space, with the constraint set by the data in an orthogonal direction.

It appears to the reviewer that the optimal size of the grids can be influenced by several inter-playing factors, e.g. the speed of particle movement, the computational workload, the tolerance of the uncertainty in the final reconstruction. It will be useful to see some discussion along this line.

2) Will a moving window approach be valuable in the data evaluation? Say, instead of dividing the image into several grids, evaluating the data within a window that is scanned with a step size that is smaller than the window itself.

3) With the proposed 3 orthogonal projections approach, there will be redundant information in the data. It is actually already there, although the 3 sets of orthogonal projection movies are not acquired concurrently. How could one make use of the redundant info?

4) As a suggestion for the future experiments, the mixture of different grains can be considered. The grains with different shapes and made of different materials can be loaded into the same cell. The interplay between different grains, which is even more scientifically interesting, can be studied.

Response to reviewer 1's comments

1) For a journal like Nat Comm, novelty is important. The authors note that an analogous method has been developed already to visualize flows in fluids. What precisely are the novel elements of the current approach? Is this the very first time it has been used for granular media? Also, they say they offer “constructive, fast velocimetry” as the benefit of this approach over existing approaches. What does this mean? It sounds like the novelty is algorithmic, chiefly. In which case, more emphasis must be designated to the methods section.

We concede that the previous phrasing was slightly misleading and didn't make the distinctions between existing methods and this work particularly clear. The previous publications also describe a correlation-based X-ray method, and thus we wanted to credit their influence on the current work. However, these publications lack essential details about their approach, and consequently we decided to pursue an independent path. With reference to the reviewer's succinct summary of the method, we believe that steps (i) and (ii) are the same as the previous work (although the first is common to many imaging methods). Step (iii) is inspired by the convolution relationship (equation (4) in the methods), which is also used in the previous work, although, to the best of our knowledge, the authors do not compute the deconvolution explicitly. Steps (iv) and (v) are completely novel and represent a significant proportion of the overall process. The comment about “fast, constructive” velocimetry was in reference to the last stage, where the velocity field is built deterministically without relying on a computationally expensive black-box optimisation procedure. One advantage of our method is that it does not impose restrictions on the velocity field *a priori*, which we believe the previous authors do, for example by imposing incompressibility. As the reviewer pointed out, this is indeed the first time any 3D X-ray velocity reconstruction has been applied to granular media, where we cannot enforce such restrictions. We have reworded the relevant paragraph in the main text (line 61) to read:

“When taken in isolation, a single set of radiographs cannot be used to recover out-of-plane velocities. However, by considering whole cross-correlation functions as opposed to only the peak locations, one can infer additional details about the out-of-plane variation, for example by assuming these cross-correlations represent the probability distribution of velocities through the beam (Fouras, Dusting, Lewis & Hourigan 2007). Under certain flow assumptions, this extra information alone is sufficient to reconstruct full three-dimensional fields, but in general it must be combined with results from other scanning angles to build a complete picture. Such an approach has been successfully applied to classical fluids using nine projection angles (Dubsky, Jamison, Irvine, Siu, Hourigan & Fouras 2010) and, by exploiting incompressibility, extended to simultaneously measure flow geometries and internal velocities with as few as three projections (Dubsky, Jamison, Higgins, Siu, Hourigan & Fouras 2012). However, these methods rely on a rigid optimisation procedure that restricts the solution to pre-assumed forms of velocity fields, for example divergence-free flows. In this paper we develop a new velocimetry method that requires no such restrictions and is thus completely general. It is applied to measure, for

the first time, three-dimensional velocity fields in granular materials, where compressibility cannot be neglected. As for the previous correlation-based methods, the new velocity reconstruction technique does not require any tomographic density maps, and we thus call it “X-ray rheography” to distinguish from existing X-ray CT technology. Full details of the approach are given in the methods and supplementary materials section.”

In addition, we believe that steps (iv) and (v) in the new method are much more generally applicable. One can conceive other situations where the PDFs of a general scalar field are available in each interrogation window, although not necessarily through the described correlation analysis. The ‘Sudoku’ style reconstruction can then be applied to re-assemble that field in three dimensions. For example, in the work of Guillard, Marks & Einav (2017) a Fourier-based method gives energy spectrums that are related to the PDFs of mean particle sizes, so an analogous method could be used to recover the three-dimensional particle size distribution in space, assuming we are able to extract the PDFs from energy spectrums. We have stressed this generality in the penultimate paragraph of the main text (line 162):

“... and Fourier-based image analysis of high-speed X-rays have been used to give experimental fabric measurements of particle size, shape and alignment (Guillard et al. 2017). Although these have currently only been applied in a two-dimensional, beam-averaged sense, it may be possible to relate the energy spectrums obtained from the Fourier analysis to PDFs of the relevant fields, allowing similar three-dimensional reconstructions to the velocity field. Such unobtrusive tools will advance the study of non-spherical grains and enable investigation of other inhomogeneous granular phenomena, for example the coupling between particle size-segregation and bulk velocity fields when grains of different shapes and sizes are present.”

2) *If I understand the method correctly, a major approximation step is when the PDF's obtained from the correlations are split up into candidate displacements. This is akin to a data-coarsening. Then all these approximated displacements are sync'ed up over the whole system, which requires another approximation, whose error is attempted to be minimized in Eqs (8) and (9). The authors concede the outcome of this approach is path dependent, even if it does save orders of magnitude of compute time over a complete optimization procedure. Because of this, the authors run the algorithm many times, creating many such paths, and average the displacements from the best 50 results. How does the standard error among these 50 tests combine with the errors in the previous steps to produce error bars for this method? It would be very important to have a formula for the accuracy of this method given that approximations are made at multiple steps.*

Yes, we recognise that there are many steps in the process that each accumulate errors. In response to this and other reviewers’ comments we have added a substantial amount of new results to the “Validation using forward projections” section of the supplementary information (now renamed “First validation case: analytical velocity field”). These involve direct calculation of the errors at all stages in the process. Whilst we have stopped short of deriving an explicit formula for the errors, it gives an indication of the relative orders of magnitude, as well as some sensitivity analysis of the most relevant parameters. Of course, the exact errors will differ depending on the particular regime being investigated, but at least this provides some useful guidelines. In reference to the standard error among the different tests, since we know the exact flow in this case, as a more appropriate measure we used the error between the exact velocity field and that obtained by averaging over the

different runs. This is what is presented on supplementary figure S5, although separate calculations of the standard error show that it has a similar magnitude.

3) The validations in the supplemental materials lack a key case. The authors reproduced their experimental flow using DEM (Fig S4) and they show the velocity fields qualitatively match the outputs of the method shown in the main text. Could the authors also run a hypothetical Xray rheography of this DEM test and show the predicted velocities side-by-side with the actual? The authors do a similar analysis earlier, using a made-up exact velocity field in Fig S2, but this analysis is less convincing because the particle flow created does not necessarily follow packing constraints so it is a less realistic case.

We have added the suggested analysis to the supplementary information, generating a series of artificial radiographs from the DEM results (as shown in supplementary figure S7 and supplementary movie 1). After carrying out the full rheography reconstruction process on these images we arrive at a velocity field that is similar to the coarse-grained DEM results. We thank the reviewer for this suggestion as it strengthens the current paper and will be a useful tool for future investigations.

4) What are the limitations of this method with regard to how thick a sample can be? Clearly as the sample gets thicker, the Xray intensity readouts are more opaque and potentially less useful to extract PDF's. But perhaps the authors have a clever way to address this.

There are actually two limitations to consider when determining the maximum sample thickness for this method. As the reviewer points out, the first is associated with the finite amount of material that can be penetrated by the X-ray sources. Very thick, dense samples will stop almost all the radiation from passing through, thus making it difficult to extract any useful information. This issue is not unique to rheography, and will have an effect on all X-ray based methods. In our particular experiment we were able to see through a 100 mm thick sample of glass beads, plus the confining container, whilst operating well within the source's energy limits (used 150 keV, maximum energy is 225 keV). The second limitation arises because the image analysis relies on the radiographs having sufficient fluctuations, or texture, to correlate between frames. This texture originates from the finite size of grains, which leads to non-uniform density fields, and subsequently intensity fields, when relatively few grains are present. When the X-ray beam passes through many more particles, each individual grain contributes proportionally less and these fields appear more homogeneous, making image correlation harder. For our system, whilst the experimental images look slightly under-exposed compared to the artificial radiographs (supplementary figure S7), the intensity fluctuations (compared to the baseline values) are the most important. When using 16-bit experimental radiographs, these fluctuations are clearly measurable. We anticipate that much thicker flows could be investigated, and ongoing work will try to establish precise limits. We have added the following comment to the main text methods section (line 198):

“This highlights the intensity fluctuations, or equivalently density fluctuations, from the baseline values, which are significant in our system due to the relatively large particle size compared to the overall system. With increasing number of particles the radiographs will become more homogeneous and the subsequent analysis will be less effective.”

Response to reviewer 2's comments

1) *The authors divided the projection images into a number of grids and performed correlation analysis to extract the amount of translational offset within each grid after a certain time delay. After the overall information in each grid (the displacement's probability distribution) is recovered, they numerically optimize the velocity vector distribution over the sub-grid pixels in the 3D space, with the constraint set by the data in an orthogonal direction. It appears to the reviewer that the optimal size of the grids can be influenced by several inter-playing factors, e.g. the speed of particle movement, the computational workload, the tolerance of the uncertainty in the final reconstruction. It will be useful to see some discussion along this line.*

Indeed, the size of the grid will have an effect on the overall results and performance of the method. Regular optical PIV also suffers from such sensitivity, and previous researchers (Stanier, Blaber, Take & White 2016; Sarno, Carravetta, Tai, Martino, Papa & Kuo 2018) have carefully investigated the effect and optimized for granular systems. We used these guidelines as a starting point for our PIV-derived method, but also investigated some of the controlling factors in more detail in the supplementary information, which has been expanded considerably to quantify the errors introduced at different stages in the process. We have included some discussion about optimal parameter choice at several points in the supplementary information, for example from line 56:

“Figure S2a shows how the size of the interrogation window influences the mean error, with larger boxes incorporating more particles and thus improving the statistical measure of velocity. However, using very large window sizes will reduce the spatial resolution of the final solution, meaning the optimal size will be a trade-off between these competing factors. The effect of the velocity magnitude, u_0 in equation (3), is displayed on figure S2b. Here we see that larger displacements lead to less accurate deconvolutions, because it becomes more difficult to correlate individual particles between images.”

2) *Will a moving window approach be valuable in the data evaluation? Say, instead of dividing the image into several grids, evaluating the data within a window that is scanned with a step size that is smaller than the window itself.*

This is an interesting suggestion which can be considered on two fronts. On the one hand, it is quite straightforward to use overlapping interrogation windows when calculating the auto- and cross-correlation functions (and subsequently velocity PDFs). We are aware this is commonly adopted in regular PIV to improve accuracy, and we actually already employ it in this paper (see main text line 203). However, after this first stage we neglect the fact that there is an overlap and consider each PDF to be associated with a cell-centre only. We then proceed with the PDF discretisation and velocity reconstruction as described to assemble the final velocities in each voxel. It might also be interesting to carry the overlapping cells through the whole process, allowing each PDF to contribute to multiple voxels (in the in-plane, as opposed to out-of-plane, directions) and formulate a new optimisation procedure. We felt that this would become too complicated at this stage in the development of the method, but have added some discussion in the main text methods section (line 203):

“Note that such an overlap is only used in the first stage of the reconstruction process, where it is employed, as for regular PIV, to improve spatial resolution and accuracy.

In the later stages we neglect the overlapping nature of the windows, considering all information to be associated with the cell centre only. It may be interesting in the future to carry the overlapping cells through the whole process, formulating new discretisation and optimisation procedures that take this extra detail into account.”

3) With the proposed 3 orthogonal projections approach, there will be redundant information in the data. It is actually already there, although the 3 sets of orthogonal projection movies are not acquired concurrently. How could one make use of the redundant info?

Actually, we are already using all of the information available. Assuming that each image is split into N^2 interrogation windows (where N is necessarily much smaller than the number of pixels), the deconvolution and discretisation processes are then used to extract candidate velocity arrays of size $N \times N \times N$ for each scanning angle and in-plane component. This gives a total of $6N^3$ pieces of information (two in-plane directions for each of the three angles). The final velocity field has $3N^3$ unknowns (N^3 for each component). Therefore there is twice the amount of information required for the process, but we use it all in the reconstruction. We have included some extra description in the methods section to clarify, firstly on line 237:

“Given N^2 interrogation windows for a single scanning direction, with each window associated with two distributions of in-plane velocity components, a single scanning direction therefore gives two $N \times N$ arrays of PDFs.”

and then on line 245:

“For each angle this process gives three-dimensional candidate arrays of size $N \times N \times N$ for the two in-plane displacement components. These constitute the displacements at different out-of-plane positions but, when taken in isolation, provide insufficient information about precisely where each displacement takes place along the beam. Taking one velocity component at a time and combining with the candidate arrays from a second scanning direction provides this extra information. Specifically, for one velocity component the discretisation process produces $2N^3$ pieces of data that are used to reconstruct the N^3 unknowns. For all three velocity components, our method produces $6N^3$ pieces of data that are used to reconstruct $3N^3$ unknowns.”

4) As a suggestion for the future experiments, the mixture of different grains can be considered. The grains with different shapes and made of different materials can be loaded into the same cell. The interplay between different grains, which is even more scientifically interesting, can be studied.

Indeed, this is one of the key areas that we would like to investigate in the future, given that all the authors have previously worked on particle segregation. This technology has the potential to shed light both through measurements of the velocity field (as described in this paper), and using an analogous reconstructive method for particle size or alignment, which would allow us to study the coupled behaviour. We have changed the penultimate paragraph of the main text (line 162) to stress these potential applications:

“... and Fourier-based image analysis of high-speed X-rays have been used to give experimental fabric measurements of particle size, shape and alignment (Guillard et al. 2017). Although these have currently only been applied in a two-dimensional, beam-averaged sense, it may be possible to relate the energy spectrums obtained from the

Fourier analysis to PDFs of the relevant fields, allowing similar three-dimensional reconstructions to the velocity field. Such unobtrusive tools will advance the study of non-spherical grains and enable investigation of other inhomogeneous granular phenomena, for example the coupling between particle size-segregation and bulk velocity fields when grains of different shapes and sizes are present.”

REFERENCES

- DUBSKY, S., JAMISON, R. A., HIGGINS, S. P A, SIU, K. K W, HOURIGAN, K. & FOURAS, A. 2012 Computed tomographic X-ray velocimetry for simultaneous 3D measurement of velocity and geometry in opaque vessels. *Exp. Fluids* **52** (3), 543–554.
- DUBSKY, S., JAMISON, R. A., IRVINE, S. C., SIU, K. K. W., HOURIGAN, K. & FOURAS, A. 2010 Computed tomographic x-ray velocimetry. *Appl. Phys. Lett.* **96** (2), 023702.
- FOURAS, A., DUSTING, J., LEWIS, R. & HOURIGAN, K. 2007 Three-dimensional synchrotron x-ray particle image velocimetry. *J. Appl. Phys.* **102** (6), 064916.
- GUILLARD, F., MARKS, B. & EINAV, I. 2017 Dynamic X-ray radiography reveals particle size and shape orientation fields during granular flow. *Sci. Rep.* **7**, 8155.
- SARNO, L., CARRAVETTA, A., TAI, Y.-C., MARTINO, R., PAPA, M.N. & KUO, C.-Y. 2018 Measuring the velocity fields of granular flows Employment of a multi-pass two-dimensional particle image velocimetry (2D-PIV) approach. *Adv. Powder Technol.* (August).
- STANIER, S.A., BLABER, J., TAKE, W.A. & WHITE, D.J. 2016 Improved image-based deformation measurement for geotechnical applications. *Can. Geotech. J.* **53** (5), 727–739.

REVIEWERS' COMMENTS:

Reviewer #1 (Remarks to the Author):

I am happy with the revisions the authors have made. This is a very nice contribution and should be published in Nature Communications.

Reviewer #2 (Remarks to the Author):

The authors have made substantial revisions in the current version of the manuscript. My comments and concerns are well addressed. I would recommend the publication of this manuscript in Nature Communications.